# Characterisation and microbial community analysis of lipid utilising microorganisms for biogas formation

**Alexis Nzila**[1]*, **Shaikh Abdur Razzak**[2], **Saravanan Sankara**[1], **Mazen K. Nazal**[3], **Marwan Al-Momani**[4], **Gi-Ung Kang**[5], **Jerald Conrad Ibal**[5], **Jae-Ho Shin**[5]

1 Department of Life Sciences, King Fahd University of Petroleum and Minerals, Dhahran, Saudi Arabia, 2 Departments of Chemical Engineering, King Fahd University of Petroleum and Minerals, Dhahran, Saudi Arabia, 3 Research Institute, Center for Environment and Water, King Fahd University of Petroleum and Minerals, Dhahran, Saudi Arabia, 4 Departments of Mathematics & Statistics, King Fahd University of Petroleum and Minerals, Dhahran, Saudi Arabia, 5 School of Applied Biosciences, College of Agriculture and Life Sciences, Kyungpook National University, Daegu, Republic of Korea

* alexisnzila@kfupm.edu.sa

**Data Availability Statement:** The data for microbial analysis can be found via the following URL: http://www.ncbi.nlm.nih.gov/bioproject/

## Abstract

In the anaerobic process, fat-oil-grease (FOG) is hydrolysed to long-chain fatty acids (LCFAs) and glycerol (GLYC), which are then used as substrates to produce biogas. The increase in FOG and LCFAs inhibits methanogenesis, and so far, most work investigating this inhibition has been carried out when FOG or LCFAs were used as co-substrates. In the current work, the inhibition of methanogenesis by FOG, LCFAs and GLYC was investigated when used as sole substrates. To gain more insight on the dynamics of this process, the change of microbial community was analysed using 16S rRNA gene amplicon sequencing. The results indicate that, as the concentrations of cooking olive oil (CO, which represents FOG) and LCFAs increase, methanogenesis is inhibited. For instance, at 0.01 g. L$^{-1}$ of FOG, the rate of biogas formation was around 8 ml.L$^{-1}$.day$^{-1}$, and this decreased to <4 ml.L$^{-1}$.day$^{-1}$ at 40 g.L$^{-1}$. Similar results were observed with the use of LCFAs. However, GLYC concentrations up to 100g.L$^{-1}$ did not affect the rate of biogas formation. Acidic pH, temperature > = 45 °C and NaCl > 3% led to a significant decrease in the rate of biogas formation. Microbial community analyses were carried out from samples from 3 different bioreactors (CO, OLEI and GLYC), on day 1, 5 and 15. In each bioreactor, microbial communities were dominated by *Proteobacteria*, *Firmicutes* and *Bacteroidetes* phyla. The most important families were *Enterobacteriaceae*, *Pseudomonadaceae* and *Shewanellaceae (Proteobacteria phylum)*, *Clostridiacea* and *Ruminococcaceae (Firmicutes)* and *Porphyromonadaceae* and *Bacteroidaceae (Bacteroidetes)*. In CO bioreactor, *Proteobacteria* bacteria decreased over time, while those of OLEI and GLYC bioreactors increased. A more pronounced increase in *Bacteroidetes* and *Firmicutes* were observed in CO bioreactor. The methanogenic archaea *Methanobacteriaceae* and *Methanocorpusculaceae* were identified. This analysis has shown that a set of microbial population is selected as a function of the substrate.

554546. Methanogenic data can be found as Supporting Information.

**Funding:** This work was funded by the Deanship of Scientific Research (DSR) of the King Fahd University of Petroleum and Minerals (KFUPM), grant IN131051 (www.kfupm.edu.sa/deanships/dsr/en/Pages/default.aspx) to AN. The funder had no role in study design, data collection and analysis, decision to publish, or preparation of the manuscript.

**Competing interests:** The authors have declared that no competing interests exist.

## Introduction

In anaerobic digestion, fat-oil-grease (FOG), which are ester of glycerol (GLYC) and long chain fatty acids (LCFAs), are first hydrolysed to LCFAs and GLYC, and the latter are converted to volatile fatty acids (VFAs) [primarily to acetate and propionate] and to $H_2$, by syntrophic acetogenic microorganisms [1,2,3]. In the last step, both acetate and $H_2$ will then be used to generate biogas (which consists of biomethane and carbon dioxide), through 2 processes, by archaea methanogens. The first involves the utilisation of acetate to produce biomethane by acetoclastic methanogens, and an important group of such archaea belong to the *Methanosaeta* genus. In the second process, $H_2/CO_2$ are converted to biomethane by hydrogenothropic archaea, among them, are *Methanosarcina*, *Methanococcus*, *Methanospirullum* genera [1,2,3]. The efficiency in biomethane production is the result a close interaction between these various microorganisms. Indeed, these microorganisms work in syntrophy, the product of one group of microorganisms is the substrate of the other, and failure to maintain an appropriate balance between them is associated with the decrease or even inhibition of biogas production [3]. The development of high through put sequencing approaches have permitted to investigate the dynamics or change of these microbial communities in the context of biogas formation [4]. More specifically, in the context of lipid as substrate (feedstock), He et al. reported a study on the microbial community succession during semi-continuous anaerobic digestion of waste cooking oil [5]. A study on microbial community changes as the results of the addition of LCFA pulses in biogas reactors was also reported [6]. Prior to the development of high throughput approaches, the technique based on denaturing gradient gel electrophoresis profiling (DGGE) was used to study changes in microbial composition caused by LCFA in relation with biogas formation [7,8]. This paper also summarises the work on the effect of FOG (representing by cooking olive oil [CO]), the LCFA oleic acid (OLEI) and GLYC, on the change of microbial community in the context of biogas formation, using 16S rRNA gene amplicon sequencing.

LCFAs have higher theoretical methane yield compared to carbohydrates and protein. For instance, a study showed that 1 g of the OLEI can yield 1 L of methane, compared to 0.35 L only for 1g of the carbohydrate glucose [9]. Thus, lipids or FOG are considered to be promising substrates in anaerobic digestion [10]. However, the utilisation of FOG in anaerobic digestion is associated with some limitations. Indeed, high concentrations of LCFAs inhibit biomethane production, as the results of microbial inhibition [10]. For instance, an early study indicated that 50% inhibition of biomethane production when 4–10 mM of Caprylic, Lauric, Myristic, Oleic acids were used [11];a range of 0.5–1 g $L^{-1}$ of OLEI and stearic acids (STEA) was associated with complete inhibition of biogas formation [12]. Similar results were reported in various studies [10,13]. It is proffered that that LCFA toxicity is due to a surfactant effect, causing the damage the cell membrane, thus microorganism death. In addition, LCFAs, through adsorption on cell membranes, reduce substrate transfer, leading to an increase in biomass flotation and washout [10,13]. However, a careful observation of the aforementioned studies shows FOG or LCFAs were primarily used as co-substrates. Thus, the second objective of this work was to investigate the inhibitor effect of FOG and the LCFAs palmitic acid (PMA), OLEI and STEA, and the alcohol GLYC, as sole substrates, on biogas formation, using inocula from a sludge of an anaerobic wastewater treatment tank. The effect of temperature, salinity and pH were also investigated on biogas formation when FOG was used as sole substrate.

Overall this work provides, for the first time, the comparative change of microbial dynamics in biogas formation with the use of FOG, LCFAs and GLYC as sole substrates. The paper also investigates the impact of increasing concentrations of these substrates on biogas formation when used as sole substrates.

## Material and methods

### Chemicals and samples

The FOG used in this study is olive cooking oil (CO) and was purchased in a commercial food supplier. The LCFAs PMA, OLEI and STEA and the chemicals used in the culture medium: $(NH4)_2SO_4$, $KH_2PO_4$, CaCl2.7H2O, MgSO4.7H2O, $Na_2HPO_4$ and FeSO4.7H2O, $Na_2S$, were purchased from Sigma-Aldrich (St. Louis, MO, USA). Inoculum samples were collected from a wastewater treatment plant, as primary sludge from anaerobic tank, of the city of Khobar, in Saudi Arabia.

### Bioreactor

Culture were carried out in 2.5 L anaerobic bioreactor (batch reactor) linked with an automatic detector of pH, and containing a temperature controller and a rotation speed adjuster. The culture consisted of $KH_2PO_4$ (1.36 g), $CaCl_2 \bullet 7H_2O$ (10.69 g), Na_2HPO4 (1.42 g) and $Na_2S$ (2g) in a total volume of 2 L. This culture was devoid of sulfate and nitrate so as to favour methanogenesis. Around 10% (v/v) of inoculum was added in the medium, along with the appropriate amount of substrate (or feedstock), as it will be specified. Thereafter, the culture was purged with $N_2$ for 15 min to creature anaerobic condition, and let to grow under agitation at speed of 50 rotation/min and at 35 $^o$C. The released biogas was monitored and collected daily by using the water displacement method, in a graduated cylinder linked to the bioreactor. At appropriate interval, around 15ml of cultures were collected for the analysis chemical oxygen demand (COD). All experiments were carried out in duplicate.

### Modelling of biogas production and initial rate of biogas production

To quantify biogas formation, the data were fitted to a Gompertz equation as follows:
$P = P_{max}*EXP[-EXP((r_m*2.718)/(P_{max}*(l-t)+1))]$ where P is the accumulated methane production (ml.L$^{-1}$ of medium), l, lag phase, expressed in day, $P_{max}$ the maximum volume of produced biogas (ml L$^{-1}$), EXP is the exponential function, and $r_m$ is the maximum biogas production rate (ml.L$^{-1}$.day$^{-1}$) and t is the time, in day [14,15]. The rate of biogas production was assessed by computing the rate of variation (in days) of the first linear phase of the biogas accumulation, as reported previously [16].

### Gas chromatography analysis (GC)

To quantify biomethane ($CH_4$), GC technique coupled with thermal conductivity detector (GC-TCD) were employed. A six ports switching valve was used to introduce the gas standards and samples, in split mode with split ratio of 18:1, onto a capillary column HP-PLOT / Q (30 m X 0.55 mm 40 um SN US9533824H), under the following oven temperature program (60 $^o$C for 2 minutes, then from 60 to 240 $^o$C with a heating rate 30 $^o$C/min, and finally an hold at 240 $^o$C for 2 minutes. The methane ($CH_4$) and carbon dioxide ($CO_2$) gases mixture was used as standards.

### Chemical oxygen Demand (COD) analysis

For COD analysis, samples was oxidised and digested by dichromate and sulfuric acid at 150˚C for 2h, and the resulting unconsumed dichromate was measured spectrophotometrically, using a spectrophotometer (DR 3900 Bench-top Spectrophotometer, Loveland, Colorado, USA) and a digital reactor (DRB200 Digital Reactor, Loveland, Colorado, USA), according to protocol established by the manufacturer.

### DNA extraction and microbial community analysis

Around 50 ml of culture samples were collected at each time point, and centrifuged at 5000 g for 5 min at 4 $^o$C. The resulting pellet, which consists of the microbial community, was preserved at 80 $^o$C until further processing. The DNA of these microorganisms were extracted using Qiagen Powerfecal Kit (Hilden, Germany), following the user's manual. The extracted DNA was used amplified the V4–V5 regions of the prokaryotic 16S rRNA genes by polymerase chain reaction (PCR) using the forward sequence 515F–GTGCCAGCMGCCGCGGTAA and the reverse sequence 907R-CCGYCAATTCMTTTRAGTTT, under the following conditions: 95˚C for 5 minutes for denaturation, followed by an initial 5 cycles of 57˚C for 30 seconds for annealing and 72˚C for extension. The initial 5 cycles were followed by another 25 cycles, at 95˚C for denaturation and 72˚C for both annealing and extension. The resulting amplified regions were sequenced using the Ion Torrent PGM (Life technologies, Carlsbad, CA) sequencing platform, using the Ion PGM™ Hi-Q™, View OT2 Kit and Ion 316™ Chip Kit V2, according to the manufacturer protocols. The raw FASTQ files were processed using QIIME (version 1.9.1) [17] and quality-filtered using Trimmomatic [18]. After removing the chimeras, the sequences were clustered into OTUs (Operational Taxonomic Units) at 97% identity using SILVA/MiDAS database (version 2.1, http://www.midasfieldguide.org).

### Statistical analyses

One-way analysis of variance (ANOVA), simple linear regression and fitting model were employed in these analyses; in addition, a pair-wise comparison Tukey's method was used. In all tests, the level of significance was p<0.05. The software MINITAB (Version16, Coventry, United Kingdom) was used in these analyses.

## Results and discussion

### Effect of CO concentration on biogas production

To assess the effect of CO on biogas formation, 2L culture were carried out in 20ml of inoculum in the presence of 40, 20, 10, 1, 0.1 and 0.01 g.L$^{-1}$ of CO. **S1 Fig** (**Supplementary material**) shows the volume of the produced biogas as a function of time, while **Fig 1** shows. Overall, the cumulated biogas volumes decrease as the concentration of CO increases (except at 0.01 g.L$^{-1}$ CO). For instance at 40 g.L$^{-1}$, the maximum cumulated volume was around 11 ml only, while at 0.1 g.L$^{-1}$ CO, this volume increased to around 33 ml. The biogas volumes corresponding to the other tested CO concentrations fell between these 2 extremes (11 and 33 ml)[**Fig 1**].

To analyse further these data, the rates of biogas formation were computed (based on the exponential phase of the Gompertz graph models from **Fig 1**) at each tested CO concentration. The results show that this inverse relationship between CO concentrations and the biogas volumes is also supported by the rate of biogas production. As the CO concentrations increase, the biogas production rates decrease, from around 8 ml.L$^{-1}$.day$^{-1}$ at 0.01 g.L$^{-1}$ CO to only 0.6 ml.L$^{-1}$.day$^{-1}$ at the highest concentration, 40 g.L$^{-1}$ CO (**Fig 2**). This decrease in biogas formation as a function of CO concentration is supported by the ANOVA test single regression (p<0.05), and within the tested CO concentrations, the correction can be predicted according to the equation: r = 6.11–0.172xC, where r is the rate of biogas formation (ml.L$^{-1}$.day$^{-1}$) and C is the CO concentration (g.L$^{-1}$). To test how the results from Gompertz model reflects the experimental data, a simple linear regression model from the origin and the ANOVA for the simple linear regression" of biogas volume data from experiments versus the Gompertz model was evaluated, and the slope parameter was tested for unity. The results of this analysis showed a significant regression model since the p-values were zero (p = 0.00), a clear indication that

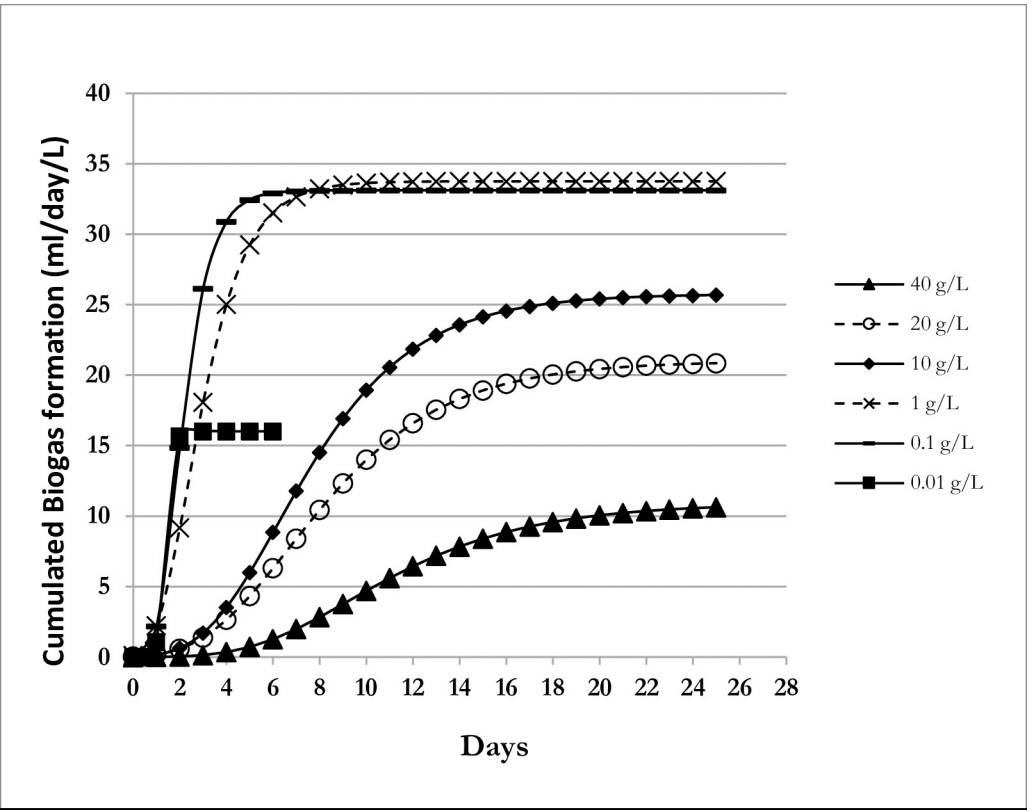

**Fig 1. Biogas formation profile from cooking oil (CO) after modeling.** Cumulated volume of biogas formation during
the anaerobic digestion of cooking oil (CO), modeled using the Gompertz equation.

the experimental and the model data do match. All these data show clearly that high CO (thus
FOG) concentrations are associated with inhibition of methanogenesis, thus, decrease of bio-
gas formation.

It is well established that the used of FOG (and also LCFAs, as it will be discussed later), is
associated with the inhibition of biogas formation [10]. However, detailed observation of pre-
vious studies shows that FOG was investigated only in the context of co-digestion (with other
substrates); therefore direct comparison of these previous data with that reported in this work
cannot be carried out. For instance, Cirne et al. investigated biomethane production in the
context of co-digestion of triolein (a FOG, an ester of OLEI and glycerol) with starch, cellulose
and whey protein [19]. Triolein concentration from 5 to 18% (w/w of COD) yielded the same
biomethane content, however, concentrations >31% inhibited methanogenesis [19]. Other
studies investigated the co-digestion of FOG waste with various substrates, and overall, the
results showed an inhibition of biomethane production with FOG >60% (of volatile solid)
[10]. In the current study, FOG was employed as sole substrate.

To ascertain that biomethane was produced during this incubation, a GC analysis of the
collected gas was carried out. The methane ($CH_4$) and carbon dioxide ($CO_2$) gas mixture was
used as standard, and the response factor of both gases were very close together with a slope
around 100 and the linearity correlation coefficient ($R_2$) were around 0.99. The results showed
that the percentage ratio between methane ($CH_4$) and carbon dioxide ($CO_2$) gases in the col-
lected sample was around 50% each. The proportion of methane in biogas is known to be vari-
able, but generally falls within the range of 40–60% for methane [2,20,21], though values as

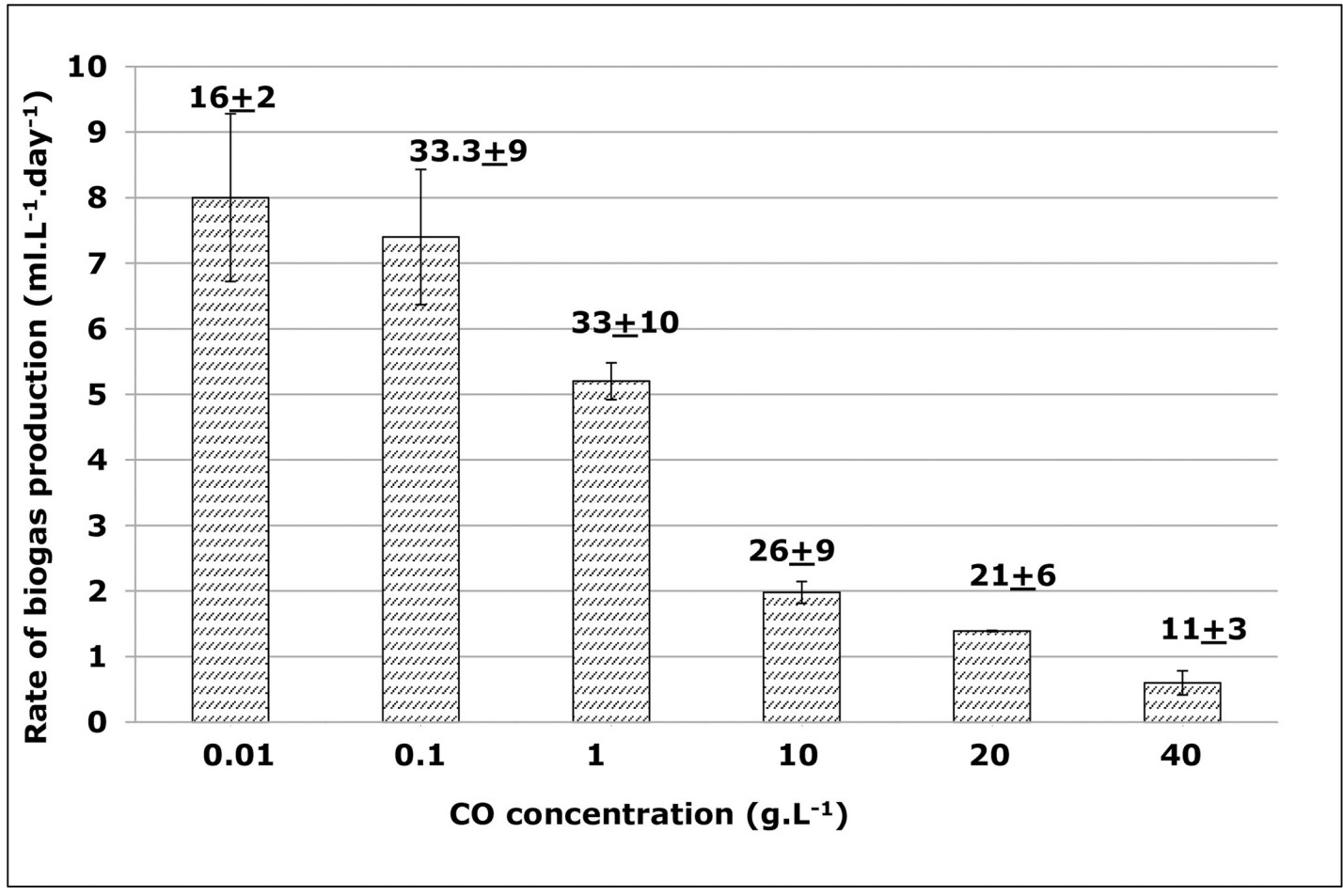

**Fig 2. Rate of biogas formation from cooking oil (CO).** Rate of the biogas formation as a function of cooking (CO) oil concentrations. These values derived from the Gompertz equation. Values on top of each column indicate the maximum accumulated biogas produced (ml.L$^{-1}$).

high as 80% can be achieved under certain conditions, for instance, high pressure [22]. Thus data reported in this work fall within this range.

Biogas is produced from the utilisation of LCFA and GLYC, as the result, the amount of carbon in the reactor will be reduced as the biogas is being formed. One way of monitoring this carbon utilisation is to quantify the chemical oxygen demand (COD) of the culture, over time. As the biogas volume increases, COD values decrease, and higher rates of reduction coincided with the exponential phase of biogas synthesis, a clear illustration of the utilisation of LCFA and GLYC by the consortium (**Fig 3**).

### Effect of temperature, pH and salinity on the formation of biogas in the presence of cooking oil (CO)

The ability of the consortium to produce biogas was assessed at 30, 35, 40 and 45 $^{o}$C; pH 4, 5, 7 and 8, and salinity 0, 1.5 and 3% NaCl. First, the culture conditions were set at 35 $^{o}$C, in the absence of salinity, and at CO of 1g.L$^{-1}$, the aforementioned pH values were tested. As Table 1 shows, both the rates of biogas formation and the cumulated maximum volumes reduced as pH values decrease. Indeed, the rates decrease from 5.2±0.3 at pH 7 to 0.56±0.1 ml.L$^{-1}$.day$^{-1}$ at pH4; values pertaining to pH5 and pH6 fell within these extremes. One way ANOVA

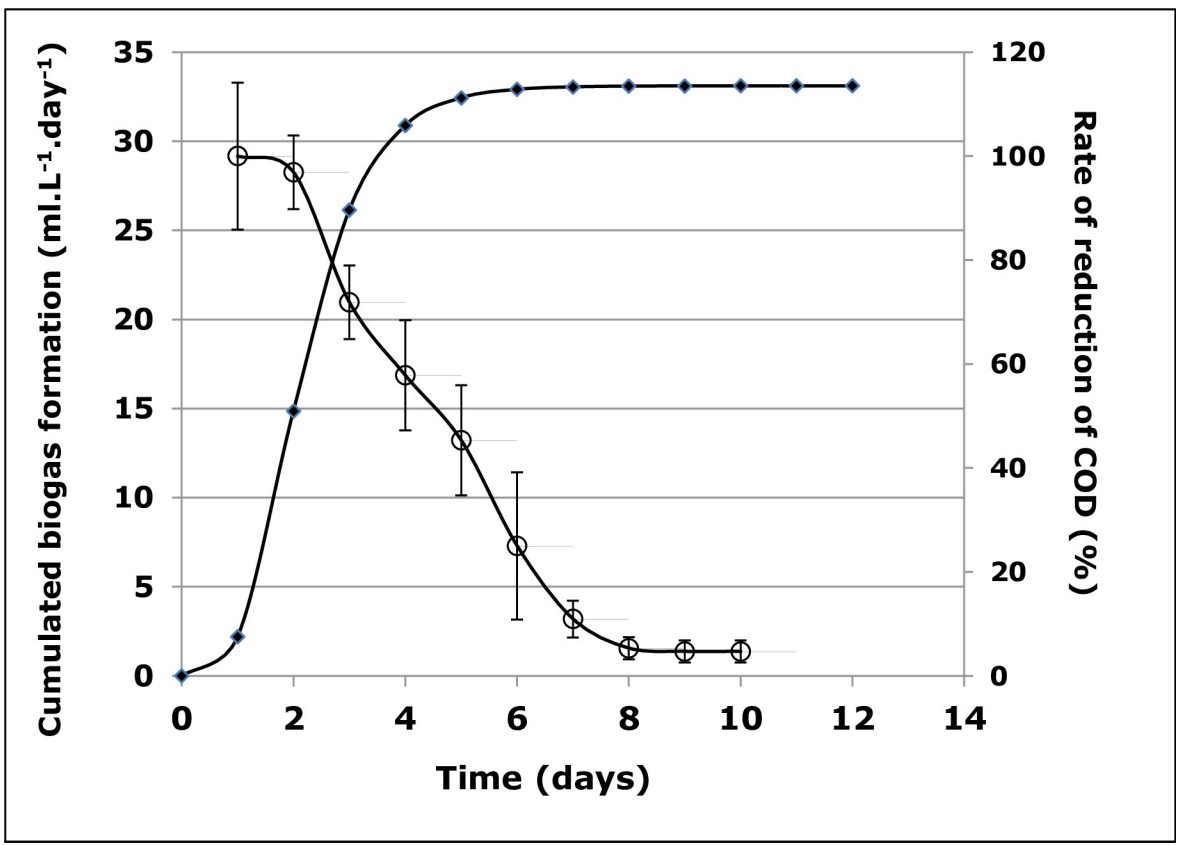

**Fig 3. Reduction rates of chemical oxygen demand (COD).** Reduction rates COD of a culture of 0.1 g L$^{-1}$ of cooking oil (CO) in relation with biogas formation. Close squares represent cumulated biogas formation (as per the Gompertz equation), while open circles represent the COD reduction.

**Table 1. Biogas formation in different conditions.** Rate of biogas formation and the maximum cumulated volumes achieved during various conditions of pH, temperature and salinity of the consortium. These values derived from the Gompertz model equations. Cooking oil (CO) was used at 1g.L$^{-1}$.

| | Conditions | Rate (ml.L$^{-1}$·day$^{-1}$) | Maximum cumulated volumes (ml) |
|---|---|---|---|
| | pH 4 | 0.56±0.1[a,b] | 8±1 |
| | pH 5 | 2±.3 [c] | 17±3 |
| pH | pH 6 | 2.96±0.8 [a,d] | 25±4.2 |
| | pH 7 | 5.2±0.3 [a, b,c,d] | 33.5±10 |
| | pH 8 | ND | ND |
| | 30 °C | 4.18±1 | 39±7 |
| Temperatures | 35 °C | 5.2±0.3 | 33.5±10 |
| | 40 °C | 5.5±0.6 | 46+8 |
| | 45 °C | ND | ND |
| | 0 | 5.2±0.3 | 33.5±10 |
| Salinity (NaCl) | 1.5% | 3.2±0.9 | 30±4 |
| | 3% | ND | ND |

a, b, c, d (pH): The difference of rates were significant ($p < 0.05$) at pH4 versus pH6 (a) pH4 versus pH7 (b); pH5 versus pH 7 (c) and pH6 versus p 7 (d).

supported these pH differences, and pairwise comparison between the rates of the different pH using Tukey's test showed significant difference between pH4 versus pH6; pH4 versus pH7; pH5 versus pH7 and pH6 versus pH7 ($p < 0.05$) [Table 1]. However, at alkaline condition (pH 8), no biogas was produced. Several reports indicate that microorganisms involved in anaerobic digestion, especially methanogens, are sensitive to pH changes. In general, optimum pH values fall between 6.5–7.5, and values below 6 and above 8 are associated with inhibitions of methanogenesis [1,21,23]. The current results are in line with these observations.

The effect of temperature at 30, 35, 40, and 45 °C was investigated by setting the conditions at pH 7 and in absence of salinity. No gas production was observed at 45 °C, and rates of biogas production fall between 4.18–5.5 and ml.L$^{-1}$.day$^{-1}$ (Table 1), while the cumulated volumes were between 33.3–46 ml. However, based on ANOVA test, the difference of biogas formation rates were not statistically significant ($p > 0.05$) [Table 1]. Temperature is another parameter that influences the biogas production. In general, increase in temperature, to a certain extent, leads to better metabolic rate of microorganisms, thus higher biogas formation. In the context of the use of lipid as foodstock, this increase in temperature will render these lipid more accessible to microorganisms, as the results of the increased of diffusion coefficients and lipid solubility in aqueous media [24]. However, evidence shows that thermophilic microorganism are more sensitive to LCFA inhibition than mesophilic ones [10,25]. In addition, high temperature tend to promote the conversion of ammonium ($NH_4^+$) to ammonia, $NH_3$, which is a toxic compound to microbes. In the current work, the increase in temperature was not associated with an increase in the rates of biogas formation. Generally the efficiency of biogas formation is associated with maintaining stable temperature in the digester [10].

In relation with salinity, 3 concentrations were tested (0, 1.5 and 3% NaCl) at 35 °C and pH 7. At 3%, no biogas was produced. The rate of biogas formation reduced from 5.2 to 3.2 ml.L$^{-1}$. day$^{-1}$ at 0 to 1.5% NaCl respectively, and a slight decrease of the maximum volume was also observed at 1.5% NaCl (**Table 1**). Thus, although these differences are not statistically significant ($p > 0.05\%$), however, these results are in line with previous work indicating salinity values $> 0.6\%$ NaCl decrease biogas formation [26].

## Effects of LFCA concentrations on biogas production

As stated earlier, FOG are hydrolysed to LCFA and GLYC before their utilisation by bacteria. Thus, to gain more insight on substrate utilisation by the consortium, biogas production was assessed using LCFAs and glycerol as sole substrates. The tested LCFAs were the saturated PMA (C16 [number of carbon]), STEA (C18) and the unsaturated OLEI (18:1 [number of carbon and number of double bond]). These LCFAs are among the most dominant long-chain fatty acids found in domestic CO [27]. The result indicate that the rates of biogas production decreased as the concentrations of the 3 tested LCFAs increased (**Fig 4**). For instance, these rates were 5.7, 4 and 2.8 ml.L$^{-1}$.day$^-$1 at 0.01, 0.1 and 1 g.L$^{-1}$ of STEA respectively. Similar range was observed with PALM (from 5.7 to 2.5 ml.L$^{-1}$.day$^-$1), while the values pertaining to the unsaturated OLEI were lower, from 2.5 to 1 ml.L$^{-1}$.day$^-$1. In comparison, at the same concentrations, CO biogas rates production were between 8–5.2 ml.L$^{-1}$.day$^-$1, values that were almost 1.5 to 2 times higher than those of LCFAs (**Fig 4**).

Several studies have been investigated the effect of LCFAs on biomethane production. However, as discussed earlier, in most of these studies, LCFAs were not used as sole substrates. For instance, a landmark study of the effect of four saturated LCFAs (caprylic [8:0], capric [10:0], lauric [12:0], and myristic [14:0] and one unsaturated OLEI [18:1] were carried out in 2.5 L bioreactor in the presence of 3 g.L$^{-1}$ of acetate, as co-substrate [11]. The results showed a 50% inhibition of biogas formation at LCFA concentrations between 0.86 and 1.44 g.L$^{-1}$ (as

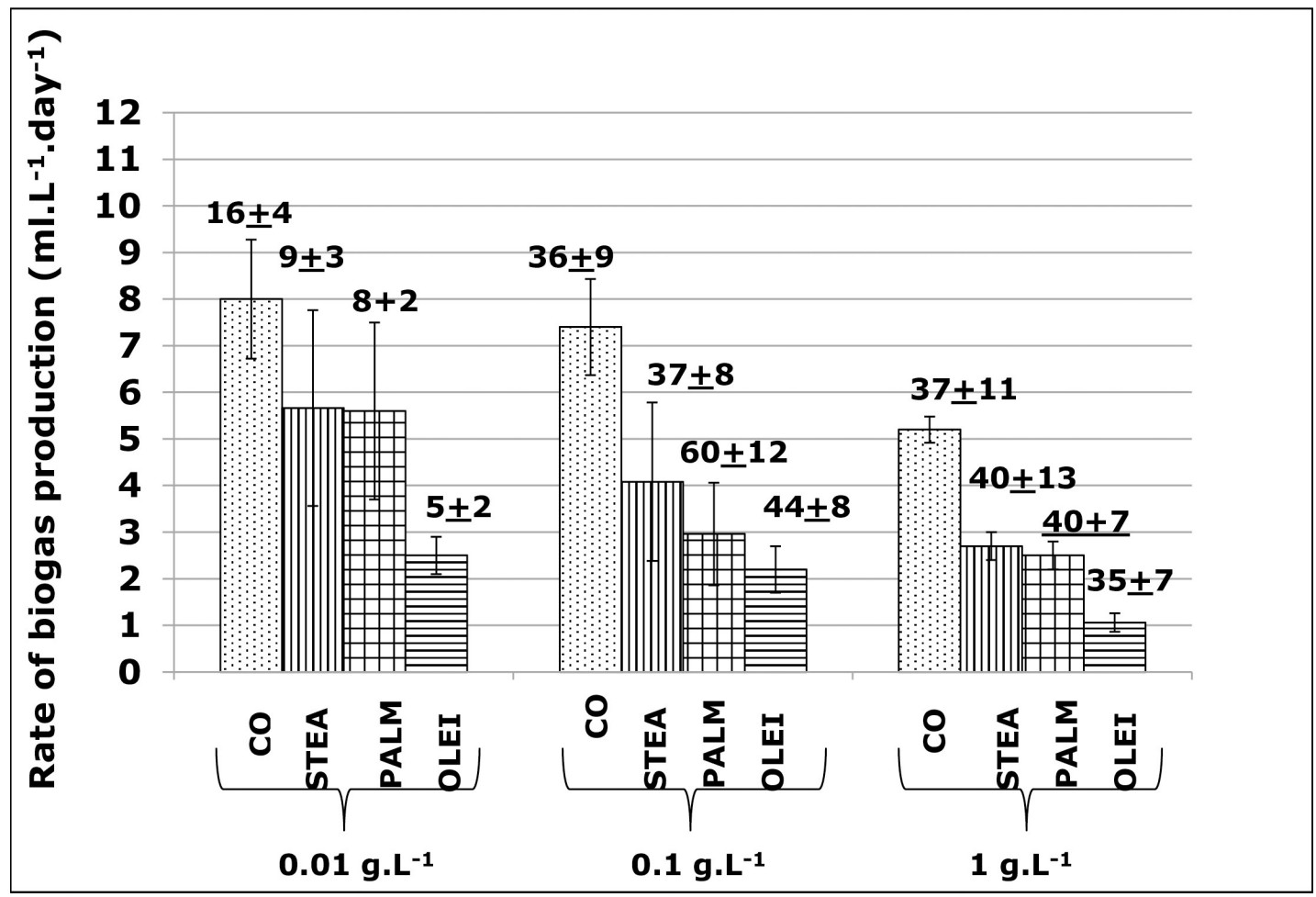

**Fig 4. Comparative rates of biogas formation between substrates.** Rate of biogas formation as a function of cooking oil (CO) and long chain fatty acids (STEA, stearic acid; PALM, palmitic acid; OLEI, oleic acid). These values derived from the Gompertz equation.

compared to the activity in the presence of acetate as sole substrate), and complete inhibition was observed with values around 2 g.L$^{-1}$ [11]. Angelika et al. carried out a similar study with OLEI and STEA, in the presence of acetate, proprionate or butyrate as co-substrates. OLEI and STEA completely inhibited methanogenesis at 0.5 and 1 g.L$^{-1}$ respectively [12]. Using different temperatures (30, 40 and 55 $^{\circ}$C), Hwu et Lettinga reported that 1 g.L$^{-1}$ of OLEI decreased methanogenesis by 50% in the presence of acetate, and this inhibition rate increased as the temperature increases [25]. In the current work, the LFCAs were used as sole substrate, therefore, a direct comparison of data cannot be made with the aforementioned studies. Nevertheless, as shown in **Fig 4**, the decrease in LCFA concentrations (1 to 0.01g g.L$^{-1}$) was associated with an increase in the rate of biogas formation, although these differences were statistically significant only between values pertaining to concentrations 1 g.L$^{-1}$ versus 0.01 g.L$^{-1}$ of LCFAs (ANOVA and Tukey's pair wise comparison, $p < 0.05$).

The current data also shows the saturated LCFAs (STEA and PALM) have higher rates of biogas formation than the un-saturated OLEI (**Fig 4**). Although these differences were statistically significant only when STEA was compared to OLEI (ANOVA and Tukey's pairwise comparison, $p < 0.05$), these results imply that OLEI was more inhibitory that the 2 saturated LCFAS. In line with these results, De Souza et al. also reported a higher inhibitory effect of

OLEI compared to PALM [28]. Likewise, using acetate as substrate, Shin et al. showed that OLEI and linoleic acid (an unsaturated acid, C18:2) were more inhibitory than saturated STEA and PALM, and this inhibitory effect increases as the number of double bond also increases [29]. This increase in LCFAs cell toxicity as a function of the number of double bonds were also reported elsewhere [11,12,30]. The mechanisms of this selective toxicity is not well known, however it was proffered that the high fluidity of unsaturated LCFAs might be one possible reason. Indeed, the melting temperatures of saturated LCFAs are generally higher than those of unsaturated ones, as the result, in the same condition, unsaturated acids have a higher fluidity, therefore more transfer to- or contact with- microorganisms, leading to her higher toxicity [30].

The current work has showed a trend towards a decrease in biogas formation rates with the 3 tested LCFAs compared to FOG (ANOVA and Tukey's pairwise comparison, p<0.05) (**Fig 4**), implying a higher inhibitory effect of LCFAs than FOG. This observation has already been documented [12]. The toxicity of LCFAs, at least partly, results from sorption or adherence on the surface of bacterial cell walls, impending the transfer of nutrients and causing the damage the cell; the free carboxylic group of LCFAs is involved in this adherence. FOG have to be hydrolysed to LCFAs first, therefore the free carboxylic groups are not readily available when FOG are used, explaining the higher inhibitory effect of LCFAs [12,15,31,32].

## Effect of GLYC concentration on biogas production

In addition to releasing LCFAs, FOG produces GLYC, which is also used as substrate for biogas formation. Thus, the effect of glycerol concentration on biogas formation was investigated, at concentration ranging from 0.01 to 100 g. $L^{-1}$. No biogas formation was observed at 0.01 g. $L^{-1}$, and as **Fig 5** showed, the rate of biogas formation fell between 9–12 ml. $^{-1}$.day$^{-1}$, and no decrease in rate of biogas formation was observed as GLYC concentration increases, up to 100 g. $L^{-1}$, the highest tested concentration, as supported by the single regression model analysis and the ANOVA test, p>0.05. These data are in contrast with those pertaining to FOG and LCFAS. Indeed, highest rates of biogas formation were observed at low concentrations of FOG and LCFAs (0.01–0.1 g.$L^{-1}$) yet, for GLYC, biogas production rates remained very high, even at a high GLYC concentration of 100 g.$L^{-1}$, concentration that was 100–1000 times higher than those tested for LCFAS and FOG. This clearly indicated that, in the context of the use of high concentration of FOG, GLYC (which is released from FOG) does not contribute in the inhibition profile of this lipid. Several investigations have been reported on the use of crude GLYC for biomethane formation. This crude GLYC, which is the product of biodiesel formation from trans-esterification of FOG by methanol [33], contains a lot of impurities, and is primary used as co-substrate in anaerobic digestion. Thus, like the case of FOG and LCFAs, direct comparison cannot be made with data reported in this work, nevertheless, the detailed review of the literature confirms that crude GLYC is less inhibitory than FOG or LCFA, and generally, the decrease in methanogenesis is observed with crude GLYC concentrations >6–12 g.$L^{-1}$ [34]. The current study shows that concentrations as high as 100 g.$L^{-1}$ of pure GLYC do not affect the rates of biogas formation (even when used in the context of batch reactor), an indication that high biogas yield can be achieved if impurities from crude GLYC are reduced.

## Analysis of microbial community

A total of 9 samples were analysed. They were collected on day 1, 5 and 15 for the 3 bioreactors, CO, OLEI and GLYC. Days 2–10 and 11–15 correspond to the exponential phase of- and the period of maximum cumulated- biogas formation respectively. The analysis of the 16S rRNA gene reads lead to the identification of a total of 4848 OTUs in samples, at 97% sequence

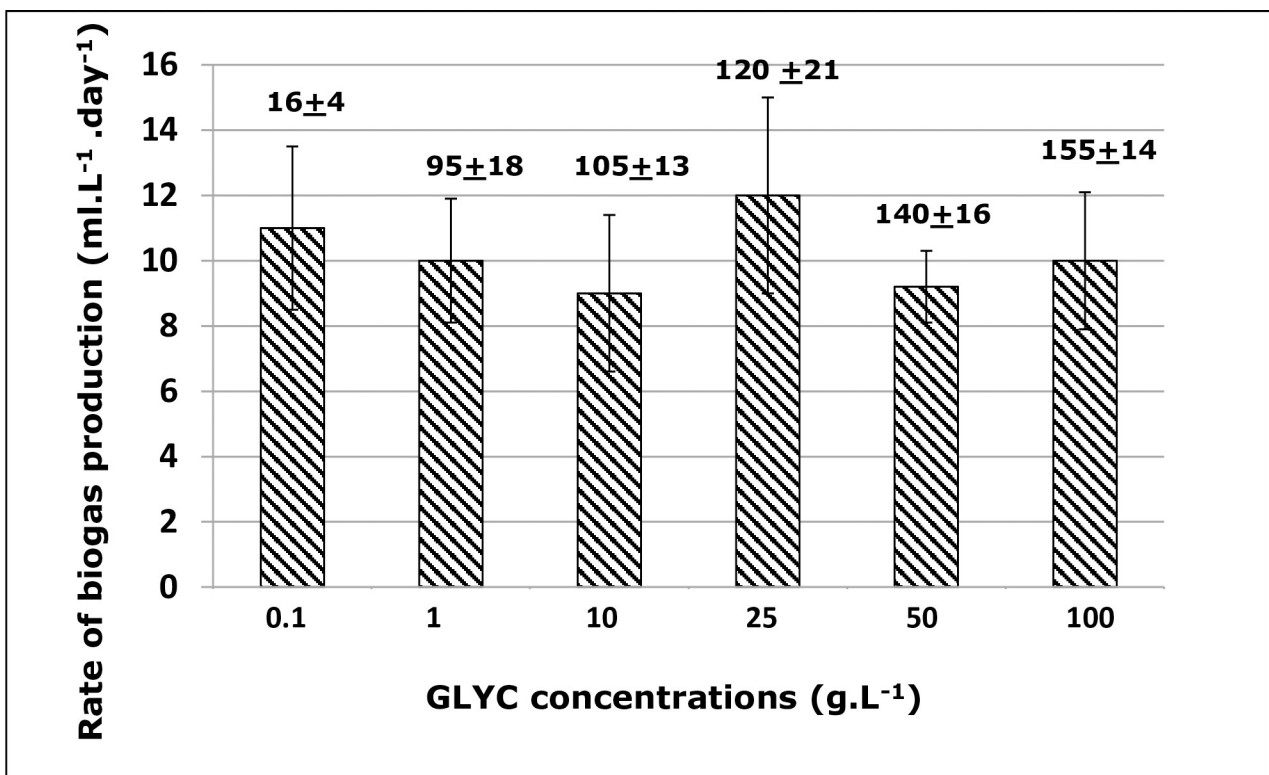

**Fig 5. Rate of biogas formation with glycerol (GLYC) as substrate.** Rate of biogas formation as a function of glycerol (GLYC) concentrations. These values derived from the Gompertz equation. Values (ml) on top of each column refer to the maximum volumes of the produced biogas.

identity. The metadata pertaining to this investigation is available at http://www.ncbi.nlm.nih.gov/bioproject/554546.

The microbial community was dominated by 3 main phyla, *Proteobacteria*, *Firmicutes* and *Bacteroidetes*, in the 3 bioreactors (**Fig 6**). Overall, *Proteobacteria* was the most dominant phylum, with the main class being *Gammaproteobacteria* (**Fig 7**), and the main families belonging to *Enterobacteriaceae*, *Pseudomonadaceae* and *Shewanellaceae* (**Fig 8**). In CO bioreactor, this phylum was high on 1 day and decreased afterwards, while in the other 2 bioreactors, this phylum remains high, even on day 5 and day 15 (during the methanogenesis phase). This is unexpected since this class of microbes decreases in anaerobic processes, because it generally consists of aerobic microbes. Its high presence in GLYC and OLEI bioreactors indicate that anaerobic conditions might have been not fully achieved, although biomethane production was observed. Similar observations have been reported in other anaerobic experiments, using various feedstock (including OLEI), in which microbial communities were reported to be relatively dominated by *Proteobacteria* or *Gammaproteobacteria*, in spite of biomethane production [35,36,37].

The family of *enterobacteriaceae (Gammaproteobacteria)* was represented by *Enterobacter*, *Raoultella*, *Citrobater* and *Klebsiella* genera (**S2 Fig, supplementary material**). Bacteria of these genera are known to be associated with fermentation or biogas formation. For instance, strains of *Enterobacter Raoultella* and *Klebsiella* have been associated with $H_2$ and biogas formation (through the generation of acetate), in bioreactors fed with wastewater or GLYC [38,39,40]. The presence of *Enterobacter* and *Citrobacter* in biogas formation has also been reported elsewhere [41,42].

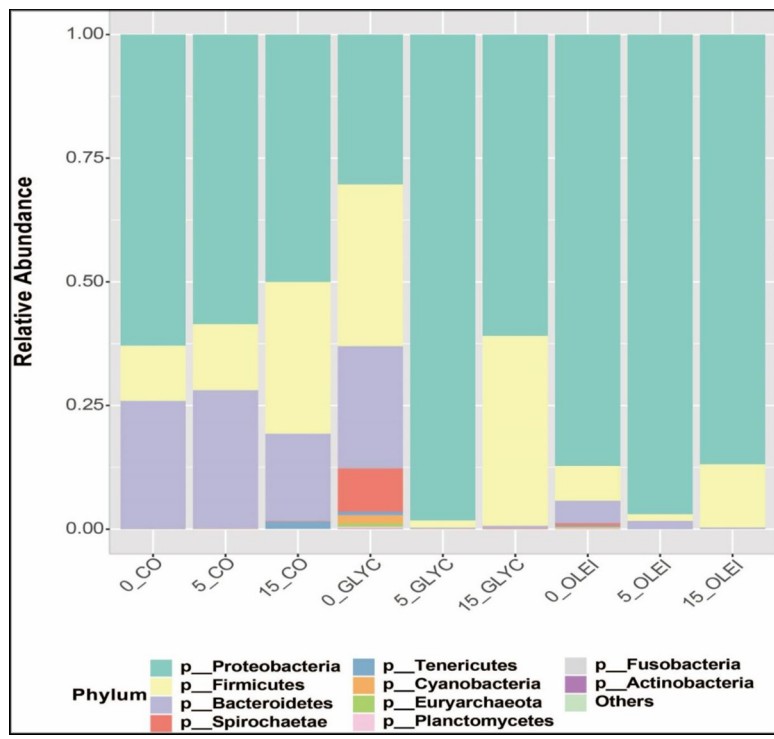

**Fig 6. Microbial abundance in relation with species phyla.** Relative abundance of microbial phyla in the presence of cooking oil (CO), glycerol (GLYC), and oleic acid (OLEI), as a function of time (1, 5, and 15 days).

Among these *Gammaproteobacteria*, the *Shewanellaceae* family (primarily of genus *Shewanella*, **S2 Fig**) forms an interesting group of bacteria. Indeed, they are known as exoelectrogenic, thus can be involved in interspecies electron transfer with methanogens, leading to an increase in the reduction of $CO_2$ to biomethane [43,44]. For instance, direct interspecies electron transfers have been demonstrated from the exoelectrogenic bacteria of *Geobacter* genus to Methanogens [43]. The presence of *Shewanellaceae* on day5 and day15 in the 3 bioreactors indicate their possible involvement in electron transfer to methanogens, thus contributing in biomethane production.

The phylum Firmicutes was the second most dominant bacteria, and consists primarily of *Clostridia* and *Bacillus* classes, and the main families were *Clostridiaceae* and *Ruminococcaceae* families (**Figs 7 and 8**). The dominant genus in this phylum was *Clostridium* (**S2 Fig**), which are obligate anaerobic. Thus, their presence is expected to be low at the early stage of the anaerobic process, but would increase with time. However, in CO bioreactor, their decrease was noticed on day 5 and day 15 compared to day 1. *Clostridium* bacteria have been associated with degradation of various organic molecules into VFA and acetate, the substrates of acetoclastic methanogens; some *Clostridium* bacteria are known to produce $H_2$, the substrate of hydrogenotrophic methanogens [26,41,45,46,47]. In addition, the predominance of these bacteria is common in anaerobic digestion, including those in which LCFAs (including OLEI) and FOG have been used as feedstock [5,8,48,49,50,51,52,53,54,55].

The phylum *Bacteroidetes*, the third most dominant, was presented by the class *Bacteroidia*, the families of *Porphyromonadaceae* and *Bacteroidaceae*, and the genus of *Bacteroides* (**Figs 6–8 and S2 Fig**). This phylum was high in CO bioreactor throughout the anaerobic process, and was less represented in GLYC and OLEI bioreactors. *Porphyromonadaceae* are syntrophic bacteria that provide acetate as substrate for methanogens [26,56,57], and the inhibition of these

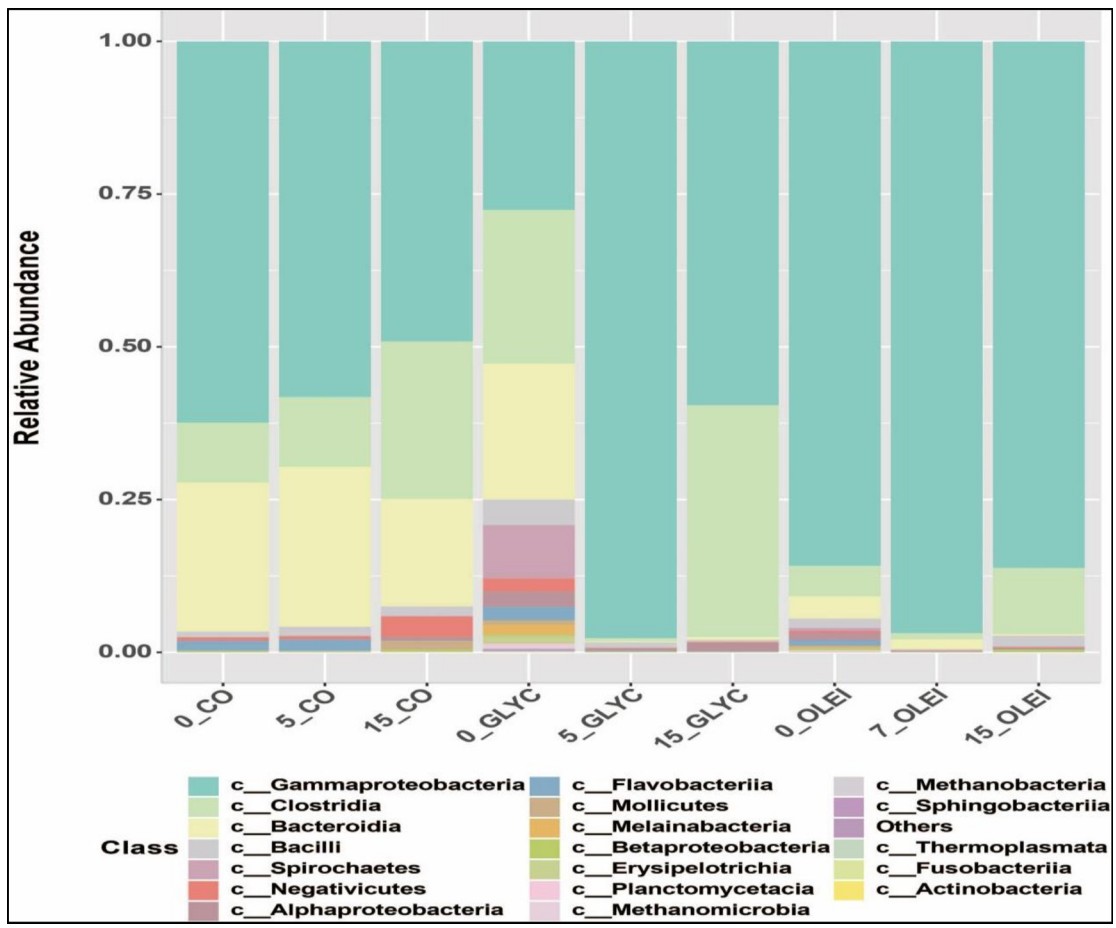

**Fig 7. Microbial abundance in relation with species classes.** Relative abundance of the class of microbes in the presence of cooking oil (CO), glycerol (GLYC), and oleic acid (OLEI), as a function of time (1, 5, and 15 days).

bacteria can result in the accumulation of LCFAs in bioreactors [58,59,60]. Thus, their presence indicates the utilisation of LCFAs. The fourth most important phylum was *Spirochaetes*, class and family of *Spirochaetes* and *Spirochaetaceae* respectively. Both *Bacteroidetes* and *Spirochaetes* have been reported in anaerobic digestion in which FOG or LFAs were used as substrates [55]

The other phyla were *Tenericutes*, *Fusobacteria* and *Actinobacteria*, which also have been reported in biogas formation [26,61]. The phyla *Cyanobacteria and Planctomycetes* were also identified, but they were primarily present the first of day of the anaerobic process.

The phylum *Euryarchaeota*, corresponding to methanogenic archaea, was identified. They belonged to 2 classes, *Methanomicrobia* and *Methanobacteria*, and the principal families were *Methanocorpusculaceae* and *Methanobacteriaceae*. In relation with the genus, *Methanocorpusculum* and *Methanobrevibacter* were the most important (**Figs 6–8** and **S2 Fig**). *Methanobrevibacter* was predominant in day 5 and 15 in CO bioreactor. The archaea *Methanobrevibacter* are known to be hydrogenotrophic, by using $CO_2$ and $H_2$ as substrates to generate biomethane [62]. Their presence has been reported in microbial communities producing biogas [63,64,65]. Some of *Methanobrevibacter* species have been shown to be acido-resistant and can grow activity in the presence of high concentration of VFA, in the context of biogas formation [66]. Since GLYC or of LFCAs lead to production of high amount of VFA, thus these acido-resistant

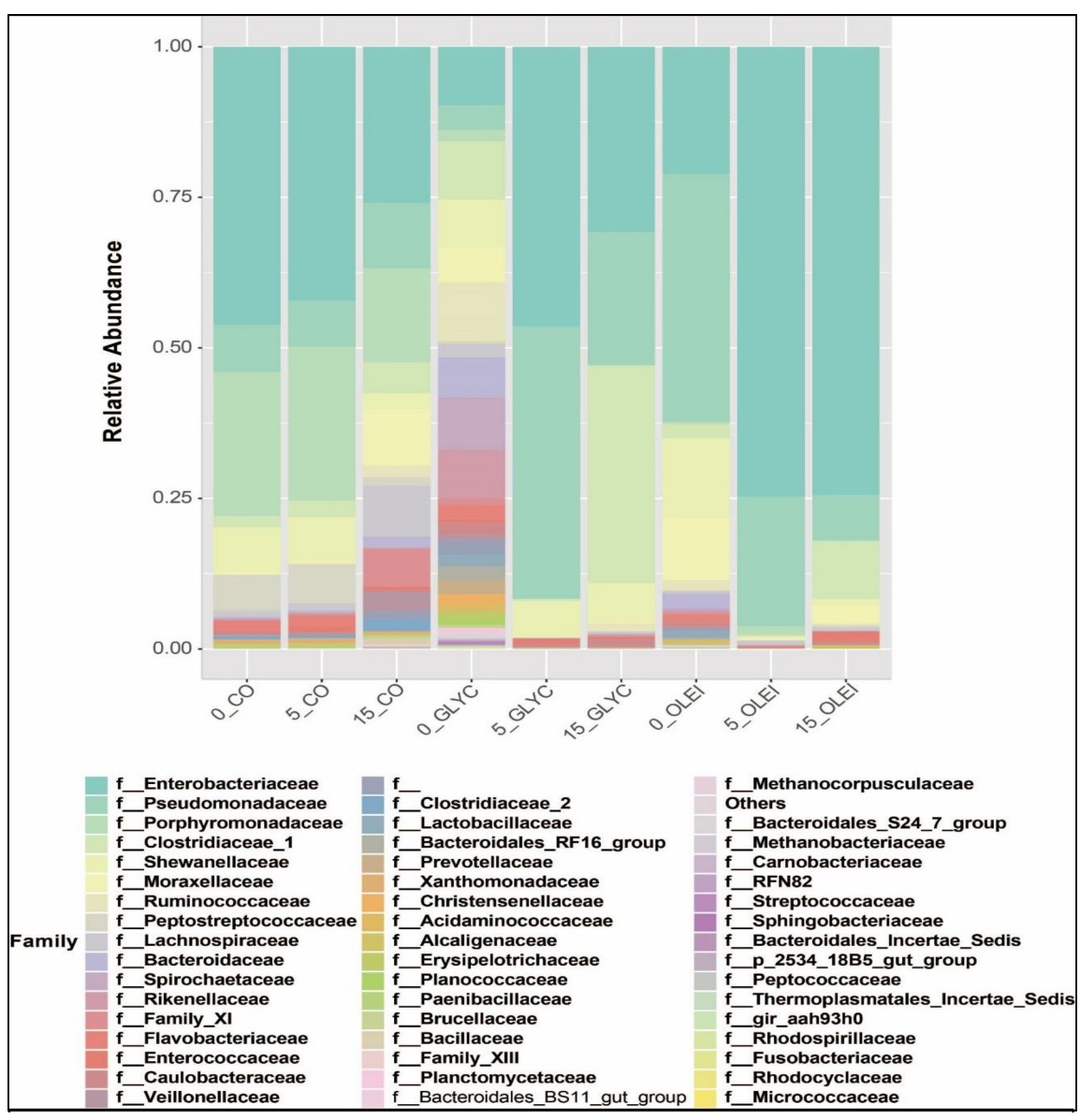

**Fig 8. Microbial abundance in relation with species families.** Relative abundance of the family of microbes in the presence of cooking oil (CO), glycerol (GLYC), and oleic acid (OLEI) as a function of time (1, 5, and 15 days).

methanogens will be favoured in these bioreactors. *Methanocorpusculum*, an archaea that can use $H_2/CO_2$ or formate as a substrate of methanogenesis, have shown to be dominant in AD processes [67,68,69,70].

CO consists of GLYC and OLEI, thus one would expect that microbial community in CO bioreactor would have a high richness than the other 2 bioreactors, since microbes specific to GLYC and OLEI degradation will both be present in CO bioreactor. This microbial community diversity was analysed through Chao1 and Shannon indices, which reflect the alpha-diversity for both richness and evenness of the OTUs. However, the results showed that no higher diversity was found in CO bioreactor, and on the contrary, a higher Chao1 value was observed in OLEI bioreactor (**Fig 9**), although these differences were not statistically significant (p>0.05). An analysis of the beta-diversity was also carried out to establish the relatedness of

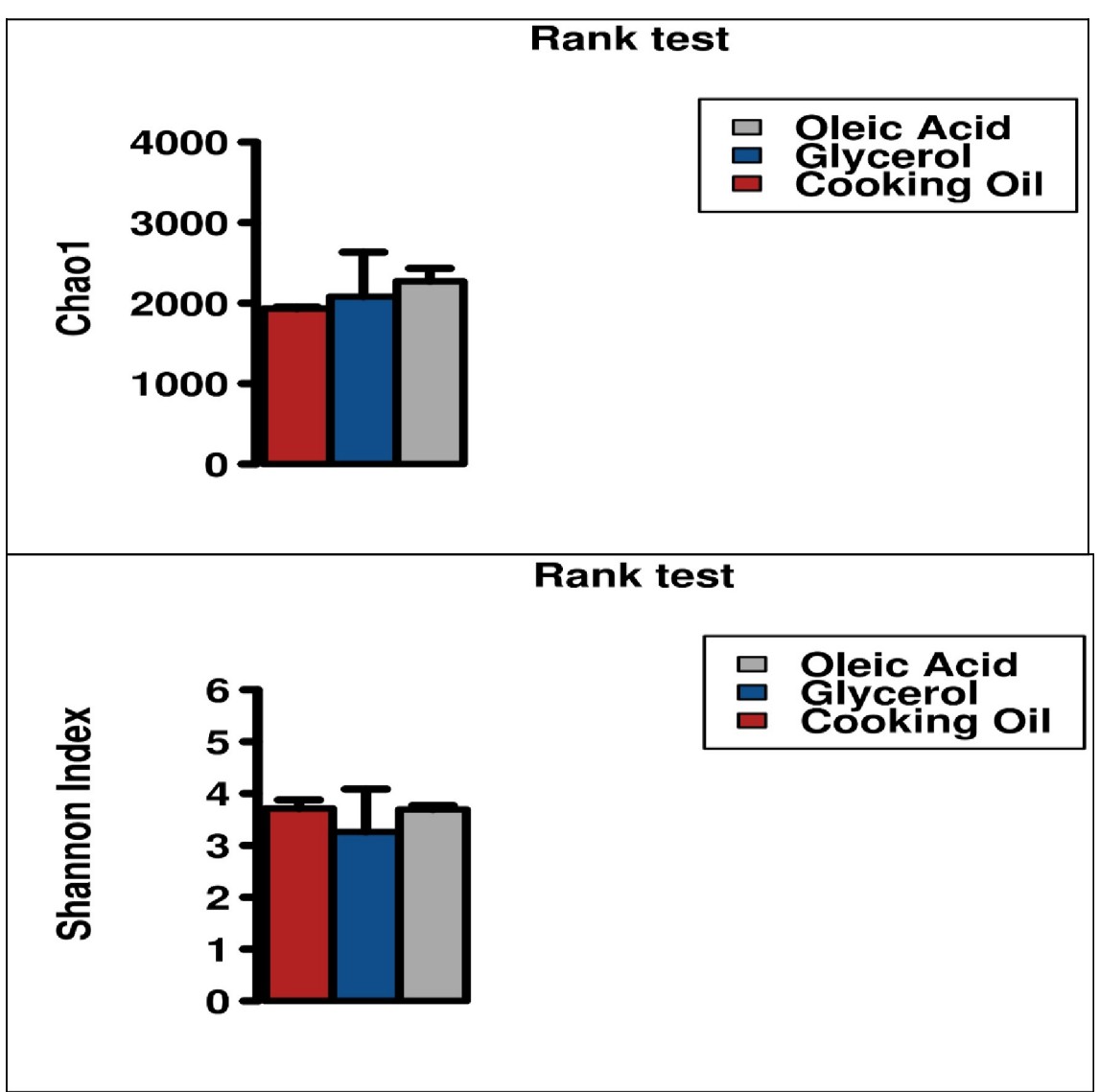

**Fig 9. Chao1 and Shannon diversity indices.** Chao1 and Shannon indices depicting the α-diversity of microbial communities in cooking oil, glycerol, oleic acid bioreactors.

microbial community in each bioreactor. The OTUs of Bray-Curtis Dissimilarity principal coordinate analysis (PCoA) (Fig 10) shows that, overall, on day5 and 15, samples of each bioreactor were closely related to each other, compared to the initial inocula. This clearly shows that each substrate selects a set of microbial community, which dominates the bioreactor on day 5 and 15. Interestingly the PCoA also showed that the GLYC and OLEI inocula were similar on day 1, but yet, as discussed, their microbial community on day5 and day15 were different, a clear illustration of selection of specific subset of microbial community based on the substrate.

Although methanogens (archaea) have been detected in this study, however their proportion is very low, in comparison with that of bacteria. Thus, the methanogen community have been underrepresented, yet biomethane have been produced in 3 bioreactors. One of the possible reasons could be that the primers used in this study did not cover sufficiently hte archaea

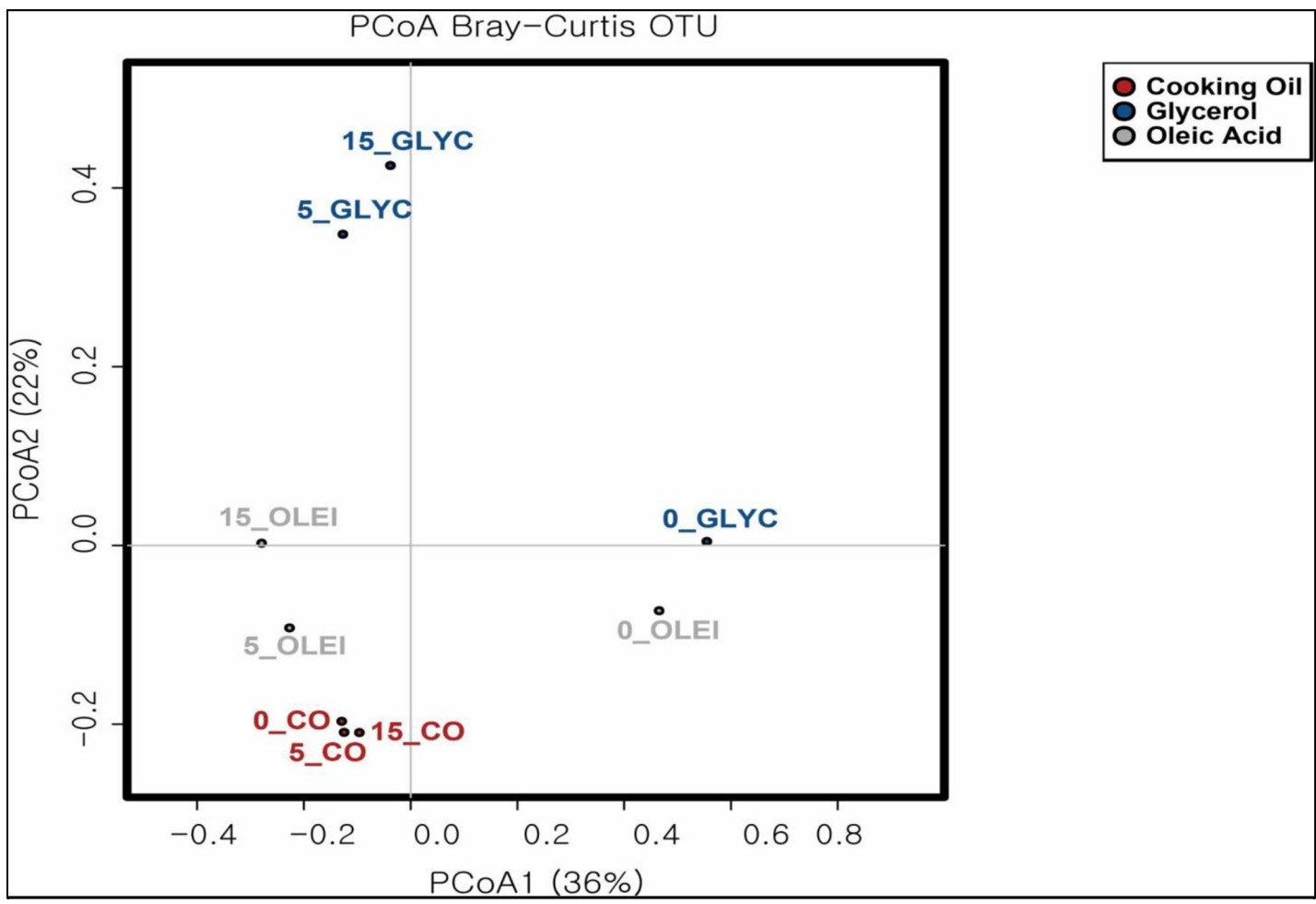

**Fig 10. Bray-Curtis principal coordinates analysis.** β-diversity of microbial community in CO, GLYC and OLEI bioreactors, as a function of time using the principal coordinates analysis of Bray-Curtis dissimilarity (PCoA).

16S rRNA genes compared to bacterial genes, leading to a higher representation of bacteria species. Thus, it is likely that other methanogen genera were present in these bioreactors.

## Conclusion

FOG, which derives from restaurants or private homes, is one of the main waste products of wastewater treatment, and therefore represents an available source of substrates for biogas formation. This work has highlighted the inhibition profile of FOG along with its 2 components, LCFAs and GLYC. Biogas formation is substantially reduced as the concentrations of LCFAs and FOG increase in the bioreactor. Overall, for the same mass, FOG (CO) was less inhibitory that the tested LCFAs, and among these LCFAs, the saturated ones were more inhibitory than the unsaturated one. Thus, the inhibition profile will also depend on the relative abundance of different types of LCFAs present in the FOG used. Interesting, GLYC, the second component of FOG, was not associated with biogas inhibition, even at a concentration as high as 100 g.L-1. These results explain, at least partly, why FOG is less inhibitory than LCFAs.

The comparative analysis of microbial communities of 3 bioreactors, each feeds with CO, OLEI or GLYC, shows that each substrate selected a specific set of microbes that efficiently

leads to biogas formation. The main dominant genus of archaea associated with biogas formation were *Methanocorpusculum and Methanobrevibacte*r.

## Supporting information

**S1 Fig. Biogas formation profile from cooking oil (CO) before modeling.** Cumulated volume of biogas formation during the anaerobic digestion of cooking oil (CO), before using the Gompertz equation model.
(TIF)

**S2 Fig. Microbial abundance in relation with species genera.** Relative abundance, as a function of time (1, 5, 15 days), of microbial genera, in the presence of cooking oil (CO), glycerol (GLYC), and oleic acid (OLEI).
(TIF)

**S3 Fig. Methanogenic data.**
(XLSX)

## Acknowledgments

The authors, AN, SS, SAR and MKN are also grateful to KFUPM for personal support.

## Author Contributions

**Conceptualization:** Alexis Nzila, Shaikh Abdur Razzak.

**Data curation:** Gi-Ung Kang, Jerald Conrad Ibal, Jae-Ho Shin.

**Formal analysis:** Alexis Nzila, Mazen K. Nazal, Marwan Al-Momani, Gi-Ung Kang, Jerald Conrad Ibal, Jae-Ho Shin.

**Funding acquisition:** Alexis Nzila, Shaikh Abdur Razzak.

**Investigation:** Saravanan Sankara, Mazen K. Nazal.

**Methodology:** Alexis Nzila, Shaikh Abdur Razzak, Saravanan Sankara, Mazen K. Nazal, Gi-Ung Kang, Jae-Ho Shin.

**Project administration:** Alexis Nzila.

**Resources:** Alexis Nzila, Shaikh Abdur Razzak, Saravanan Sankara, Jae-Ho Shin.

**Software:** Mazen K. Nazal, Marwan Al-Momani, Gi-Ung Kang, Jerald Conrad Ibal, Jae-Ho Shin.

**Supervision:** Jae-Ho Shin.

**Validation:** Alexis Nzila, Shaikh Abdur Razzak, Saravanan Sankara, Mazen K. Nazal, Gi-Ung Kang, Jerald Conrad Ibal, Jae-Ho Shin.

**Visualization:** Gi-Ung Kang, Jerald Conrad Ibal, Jae-Ho Shin.

**Writing – original draft:** Alexis Nzila, Shaikh Abdur Razzak.

**Writing – review & editing:** Alexis Nzila, Shaikh Abdur Razzak, Saravanan Sankara, Mazen K. Nazal, Marwan Al-Momani, Gi-Ung Kang, Jerald Conrad Ibal, Jae-Ho Shin.

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
