## [Decision Letter · Decision Letter 0]

27 Jun 2019

PONE-D-19-15756

Characterisation and metagenomic analysis of lipid utilising microorganisms for biogas formation.

PLOS ONE

Dear Dr. Nzila,

Thank you for submitting your manuscript to PLOS ONE. After careful consideration, we feel that it has merit but does not fully meet PLOS ONE’s publication criteria as it currently stands. Therefore, we invite you to submit a revised version of the manuscript that addresses the points raised during the review process.

THE REVIEWER RAISES A NUMBER OF MAJOR ISSUES THAT MUST BE ADDRESSED DURING REVISION. FOR INSTANCE, A BETTER DESCRIPTION OF WHAT EXACTLY WAS DONE ALONG WITH DETAILED PROTOCOLS MUST BE INCLUDED.

We would appreciate receiving your revised manuscript by Aug 11 2019 11:59PM. To enhance the reproducibility of your results, we recommend that if applicable you deposit your laboratory protocols in protocols.io, where a protocol can be assigned its own identifier (DOI) such that it can be cited independently in the future. For instructions see: http://journals.plos.org/plosone/s/submission-guidelines#loc-laboratory-protocols

We look forward to receiving your revised manuscript.

Kind regards,

Juan J Loor

Academic Editor

PLOS ONE

Journal Requirements:

Reviewers' comments:

Reviewer's Responses to Questions

**Comments to the Author**

1. Is the manuscript technically sound, and do the data support the conclusions?

Reviewer #1: Partly

2. Has the statistical analysis been performed appropriately and rigorously? 

Reviewer #1: No

3. Have the authors made all data underlying the findings in their manuscript fully available?

Reviewer #1: No

4. Is the manuscript presented in an intelligible fashion and written in standard English?

Reviewer #1: No

5. Review Comments to the Author

Reviewer #1: Review on the manuscript entitled „Characterisation and metagenomic analysis of lipid utilising microorganisms for biogas formation“ submitted to PLOS ONE (PONE-D-19-15756).

The manuscript describes anaerobic digestion experiments with fat-oil-grease as substrate, in more detail with cooking olive oil, three different long chain fatty acids and glycerol. Inhibitory of the substrates were evaluated in batch tests and the microbial communities inherent of this processes were analyzed. In contrast to glycerol which showed no inhibitory effect, the long chain fatty acids were inhibiting the biogas production even stronger than cooking oil. This was also reflecting in the different microbial community compositions.

Even though the experiments are thoroughly conducted and probably scientifically sound, I would like to raise major concerns about the way data was analyzed and presented in the manuscript.

1. There was no metagenomics done! This is misleading. Metagenomics means the sequencing and analysis of all DNA present in a community. In this manuscript actually amplicon sequencing was described, so the sequencing and analysis of a single marker gene. This needs to be corrected.

2. The methods are described to insufficient detail. I would like to highlight this with a few examples:

a. L128-139: Number of replicates was not given.

b. L152-160: What was measured? It is simply not mentioned here.

c. L172: With one-way ANOVA it can only be revealed if significant differences are present in the dataset at all. To see which ones are significantly different post hoc tests have to be applied taking into account the inflation of alpha error when doing multiple comparisons. This is not mentioned is the methods section but in Table 1 it is stated which ones are different. This can simply not be deduced from the ANOVA as described in the methods section.

d. L182: Which primers have been used? Giving the variable regions which have been sequenced is not enough.

e. L214: goodness of fit is not given but highly desired to judge the suitability of the Gompertz equation to fit the data presented in this manuscript.

3. As far as I am concerned and this is in line with a very important publication on this topic (Klindworth et al. https://www.ncbi.nlm.nih.gov/pubmed/22933715), there are no general primers which cover bacteria and archaea equally good. As the actual primers are not given I can just speculate, but primers used in this study seem to be optimized for bacteria and not for archaea because (i) the archaeal abundance is quite low and (ii) there is no acetoclastic methanogen present. Depending on the coverage of the primer for archaea (can be looked up in the mentioned publication most probably), archaeal reads should be excluded from the analysis as with these primers they cannot be sufficiently covered. This of course limits the scope of this manuscript.

4. L193: GreenGenes is quite old, last update was 2012 or 2013. Hence, it misses recently discovered or described organisms. Just as an example the important group of syntrophic acetate oxidizing bacteria is almost completely missing in this dataset. To include also latest knowledge use latest release of Silva database or the MiDAS database (http://www.midasfieldguide.org/en/download/) which was manually curated for activated sludge, anaerobic digesters and influent wastewater microbiomes. I strongly suggest re-running the analysis with one of the two mentioned databases. This might have little impact on the obtained results. But at least you can be sure to have the latest available information included. Both databases are already formatted to be readily used in Qiime.

5. Raw sequence data (fastq files) are not uploaded to a database (EMBL ENA or NCBI SRA) and hence, made publicly available. This is not in line with the PLOS data policy and data availability.

6. Spelling mistakes needs to be correctd. Here, I can just give a few examples:

a. L 50: Methanomassiliicoccaceae

b. L52: Methanobrevibacter

c. L66: archaea

d. L67: genera

e. L69: syntrophy

f. L447: Proteobacteria

7. Rudimentary headlines are given: for example: L195, L196, L325 and so on. These are no proper headlines for a scientific manuscript.

6. PLOS authors have the option to publish the peer review history of their article (what does this mean?). If published, this will include your full peer review and any attached files.

Reviewer #1: No

---

## [Author Response · Author response to Decision Letter 0]

10 Aug 2019

RESPONSE TO THE EDITOR AND REVIEWER COMMENTS: PONE-D-19-1575 

Journal Requirements:

1JR. When submitting your revision, we need you to address these additional requirements.

Our answer: We have carefully read the requirements and have made the necessary changes so that our manuscript meet the journal requirements. We have also checked all the figures in “PACE application.

2JR. In your Data Availability statement, you have not specified where the minimal data set underlying the results described in your manuscript can be found. PLOS defines a study's minimal data set as the underlying data used to reach the conclusions drawn in the manuscript and any additional data required to replicate the reported study findings in their entirety. All PLOS journals require that the minimal data set be made fully available. For more information about our data policy, please see http://journals.plos.org/plosone/s/data-availability.

Our response: We have made our data available through the addition of a Supporting information as S3_Fig_methanogenic data_excell, and the sequencing data have been made available through the NCBI site, under the following references: SUB5945674 (SubmissionID), PRJNA554546 (BioProject ID), the BioProject database (http://www.ncbi.nlm.nih.gov/bioproject/554546)

Comments to the Author

1Co_Au. Is the manuscript technically sound, and do the data support the conclusions?

Reviewer #1: Partly

Our Response: We feel that our data do support the conclusion our work, and the protocols and methodology we used have been reported elsewhere and we have quoted the appropriate references.

2Co-Au. Has the statistical analysis been performed appropriately and rigorously? 

Reviewer #1: No

Our response: We have included a statistician as a co-author, Dr Marwan Al-Momani, and he has re-analysed our data appropriately. All the changes are highlighted in yellow in the text. Thus, all the statistical tests are now sound and appropriate. 

3Co_Au. Have the authors made all data underlying the findings in their manuscript fully available?

Reviewer #1: No

Our response: This has been addressed, as mentioned in our response in 2JR

4Co_Au. Is the manuscript presented in an intelligible fashion and written in standard English?

Reviewer #1: No

 Our response: We have carefully re-read the manuscript have made the necessary changes. 

5. Review Comments to the Author

Reviewer #1: Review on the manuscript entitled „Characterisation and metagenomic analysis of lipid utilising microorganisms for biogas formation“ submitted to PLOS ONE (PONE-D-19-15756).

The manuscript describes anaerobic digestion experiments with fat-oil-grease as substrate, in more detail with cooking olive oil, three different long chain fatty acids and glycerol. Inhibitory of the substrates were evaluated in batch tests and the microbial communities inherent of this processes were analyzed. In contrast to glycerol which showed no inhibitory effect, the long chain fatty acids were inhibiting the biogas production even stronger than cooking oil. This was also reflecting in the different microbial community compositions.Even though the experiments are thoroughly conducted and probably scientifically sound, I would like to raise major concerns about the way data was analyzed and presented in the manuscript.

1Re. There was no metagenomics done! This is misleading. Metagenomics means the sequencing and analysis of all DNA present in a community. In this manuscript actually amplicon sequencing was described, so the sequencing and analysis of a single marker gene. This needs to be corrected.

Our answer: We thank the reviewer for this comment. We agree that the use of the term Metagenomics is misleading and may confuse the readers. Thus, in the text, we have changed metagenomics by “16S rRNA gene amplicon sequencing” or microbial community analysis

2 Re. The methods are described to insufficient detail. I would like to highlight this with a few examples:

Our response: We have added additional information, whenever applicable, in the Material & Methods. Specific points raised by the reviewer have been addressed below.

a. L128-139: Number of replicates was not given.

Our response: Experiments were carried out in duplicate and this has added in line 144

b. L152-160: What was measured? It is simply not mentioned here.

Our response: we thank the reviewer for this comment. Methane was measured in this experiment, and this has now been added in the text line 158. 

c. L172: With one-way ANOVA it can only be revealed if significant differences are present in the dataset at all. To see which ones are significantly different post hoc tests have to be applied taking into account the inflation of alpha error when doing multiple comparisons. This is not mentioned is the methods section but in Table 1 it is stated which ones are different. This can simply not be deduced from the ANOVA as described in the methods section.

Our response: As mentioned in our Response 2 Co-Au, we have included a statistician as a co-author, Dr Marwan Al-Momani, and he has re-analysed our data appropriately. Basically the ANOVA test has been combined with pairwise comparison, using TUKEY’s test, which is in line with the reviewer comment

d. L182: Which primers have been used? Giving the variable regions which have been sequenced is not enough.

Our answer: We used the V4-V5 region of the 16S rRNA gene using the forward sequence 515F : 5’ - GTGCCAGCMGCCGCGGTAA – 3’ and the reverse sequence 907R-MM : 5’ - CCGYCAATTCMTTTRAGTTT – 3’

e. L214: goodness of fit is not given but highly desired to judge the suitability of the Gompertz equation to fit the data presented in this manuscript.

Our response: To address this reviewer comment on similarity of experimental data versus those obtained by Gompertz equation model, we constructed a scatterplots of pairs of observations where the first coordinate is the observed experimental value, and the second coordinate is the corresponding Gompertz model value, at the same time. All graphs showed approximately a linear line from the origin. To confirm our findings, we fitted a simple linear regression model from the origin of the form Y=BX, and we tested the hypothesis B=1 versus B ≠1, and the hypothesis was not rejected in all cases, and this supports our plot’s findings. Thus, the Gompertz model equation reflects the experimental data, and this is why this model is commonly used in the biogas production. This information has been also captured in the text (see line 231). 

3 Re. As far as I am concerned and this is in line with a very important publication on this topic (Klindworth et al. https://www.ncbi.nlm.nih.gov/pubmed/22933715), there are no general primers which cover bacteria and archaea equally good. As the actual primers are not given I can just speculate, but primers used in this study seem to be optimized for bacteria and not for archaea because (i) the archaeal abundance is quite low and (ii) there is no acetoclastic methanogen present. Depending on the coverage of the primer for archaea (can be looked up in the mentioned publication most probably), archaeal reads should be excluded from the analysis as with these primers they cannot be sufficiently covered. This of course limits the scope of this manuscript.

Our answer: We thank the reviewer for this comment. To the best of our knowledge, there is no study yet combining primer sets for microbiome identification of archaeal and bacterial sequences. In the current study, we focused on identifying as many OTUs as we could, using a modified primer sequences that covers the V4-V5 region. During the modification of our primers, we opted to include as many archaeal sequences we could in the curation of the sequences.

4 Re. L193: GreenGenes is quite old, last update was 2012 or 2013. Hence, it misses recently discovered or described organisms. Just as an example the important group of syntrophic acetate oxidizing bacteria is almost completely missing in this dataset. To include also latest knowledge use latest release of Silva database or the MiDAS database (http://www.midasfieldguide.org/en/download/) which was manually curated for activated sludge, anaerobic digesters and influent wastewater microbiomes. I strongly suggest re-running the analysis with one of the two mentioned databases. This might have little impact on the obtained results. But at least you can be sure to have the latest available information included. Both databases are already formatted to be readily used in Qiime.

- Our answer: We thank the reviewer for this comments. As the reviewer has suggested, we have re-ran the analysis using the latest version of SILVA (132). We have employed SILVA databases, and we have obtained more OTUs with this SILVA software compared to the previous one, Greengenes. 

However, when comparing sample counts, the observed counts were, overall, higher with Greengenes compared to SILVA. More specifically, the nine samples (1-3 for CO), 4-6 for GLYC and 7-9 for OLEI) produced counts of 17,368; 10,906; 12,232; 14,831; 20,192; 16,362; 17,507; 17,266; 16,360 respectively with the Greengenes , while those of the SILVA were (17,159; 16,310; 15,563; 14,928; 19,609; 16,065; 16,187; 17,335; 16,374). Thus, the higher number of observed counts generally came from using the Greengenes database (only 3 samples from SILVA gave higher OTU counts compared to Greengenes). 

In relation with the alpha diversity, there was no change in shannon index, while an increase in diversity in the chao1 was observed for SILVA. 

The PCoA result from the Bray-Curtis dissimilarity showed a more scattered pattern from the initial dates of sampling from SILVA compared to Greengenes. 

The same clustering of the succeeding sampling dates were observed in both databases. 

Therefore, we wish to argue that there is no reason to change the database used in the study, since it had little impact on the result. Moreover, the syntrophic acetate oxidizing bacteria did not increase when SILVA database was used. 

5 Re. Raw sequence data (fastq files) are not uploaded to a database (EMBL ENA or NCBI SRA) and hence, made publicly available. This is not in line with the PLOS data policy and data availability.

Our answer: As mentioned earlier (our Response in 2JR), the data has now been made available on NCBI database, and can be found on the link (http://www.ncbi.nlm.nih.gov/bioproject/554546)

6 Re. Spelling mistakes needs to be correctd. Here, I can just give a few examples:

Our answer: We have carefully checked the manuscript for any mistake and have made the necessary corrections. 

a. L 50: Methanomassiliicoccaceae

Our answer: Done 

b. L52: Methanobrevibacter

Our answer: DONE

c. L66: archaea

Our answer: DONE

d. L67: genera

Our answer: DONE

e. L69: syntrophy

Our answer: DONE 

f. L447: Proteobacteria

Our answer: DONE

7 Re. Rudimentary headlines are given: for example: L195, L196, L325 and so on. These are no proper headlines for a scientific manuscript.

Our answer: Throughout the text, we have re-written these headlines (see L 202, 203, 342, 414)

 END DOCUMENT

---

## [Decision Letter · Decision Letter 1]

2 Sep 2019

PONE-D-19-15756R1

Characterisation and microbial community analysis of lipid utilising microorganisms for biogas formation.

PLOS ONE

Dear Dr. Nzila,

Thank you for submitting your manuscript to PLOS ONE. After careful consideration, we feel that it has merit but does not fully meet PLOS ONE’s publication criteria as it currently stands. Therefore, we invite you to submit a revised version of the manuscript that addresses the points raised during the review process.

We would appreciate receiving your revised manuscript by Oct 17 2019 11:59PM. To enhance the reproducibility of your results, we recommend that if applicable you deposit your laboratory protocols in protocols.io, where a protocol can be assigned its own identifier (DOI) such that it can be cited independently in the future. For instructions see: http://journals.plos.org/plosone/s/submission-guidelines#loc-laboratory-protocols

We look forward to receiving your revised manuscript.

Kind regards,

Juan J Loor

Academic Editor

PLOS ONE

Reviewers' comments:

Reviewer's Responses to Questions

**Comments to the Author**

1. If the authors have adequately addressed your comments raised in a previous round of review and you feel that this manuscript is now acceptable for publication, you may indicate that here to bypass the “Comments to the Author” section, enter your conflict of interest statement in the “Confidential to Editor” section, and submit your "Accept" recommendation.

Reviewer #1: (No Response)

2. Is the manuscript technically sound, and do the data support the conclusions?

Reviewer #1: No

3. Has the statistical analysis been performed appropriately and rigorously? 

Reviewer #1: Yes

4. Have the authors made all data underlying the findings in their manuscript fully available?

Reviewer #1: Yes

5. Is the manuscript presented in an intelligible fashion and written in standard English?

Reviewer #1: Yes

6. Review Comments to the Author

Reviewer #1: Review on the revised manuscript entitled „Characterisation and metagenomic analysis of lipid utilising microorganisms for biogas formation“ submitted to PLOS ONE (PONE-D-19-15756R1).

The authors carefully revised the manuscript and hence, improved it. Nonetheless, there are still one minor and two critical points I need to address:

1. Primer names AND sequences need to be added to the methods section of the manuscript.

2. Coverage of primers: To be very clear on this: There are no universal primers covering bacteria AND archaea satisfyingly. What is the coverage for bacteria and archaea for the primer pair used for this manuscript? To answer this either the primers have been used also in the mentioned study from Klindworth et al. (2013) and you can easily look that up or you need to re-run the analysis Anna Klindworth and colleagues did. However, this is not the point here. It is quite obvious when looking at the results that these primers are not covering the archaea well enough. Who is consuming the acetate as highly important intermediate during biogas formation? There are no acetoclastic methanogens detected and there are no syntrophic acetate oxidizers detected! Hence, it is strikingly obvious that these primers give a misleading picture of the archaeal community. Consequently, when knowing this, it is not scientifically sound to present the archaeal community composition based on these primers. Hence, archaeal reads should be filtered out and only the bacterial community should be presented. And yes, there are a lot of studies who use two separate primer sets, one for bacteria and one for archaea (either targeting archaeal 16S rRNA genes or mcrA gene for example).

3. Greengenes vs SILVA/MiDAS: I do not quite understand why the amplicon data analysis yields more counts, more OTUs when using Greengenes compared to SILVA. This does not make any sense. OTUs are constructed first and just after this step representative sequences of each OTU are taxonomically assigned by using one of the databases. Hence, OTU construction happens before the databases come into play and hence, is completely independent of the database. How then can the database influence the number of counts? And again Greengenes is outdate and should not be used anymore!

7. PLOS authors have the option to publish the peer review history of their article (what does this mean?). If published, this will include your full peer review and any attached files.

Reviewer #1: No

---

## [Author Response · Author response to Decision Letter 1]

13 Sep 2019

RESPONSE TO THE REVIEWER COMMENTS

PONE-D-19-15756R1

Characterisation and microbial community analysis of lipid utilising microorganisms for biogas formation.

1. Primer names AND sequences need to be added to the methods section of the manuscript. 

1. Our response: This has been done (page 7). 

2. Coverage of primers: To be very clear on this: There are no universal primers covering bacteria AND archaea satisfyingly. What is the coverage for bacteria and archaea for the primer pair used for this manuscript? To answer this either the primers have been used also in the mentioned study from Klindworth et al. (2013) and you can easily look that up or you need to re-run the analysis Anna Klindworth and colleagues did. However, this is not the point here. It is quite obvious when looking at the results that these primers are not covering the archaea well enough. Who is consuming the acetate as highly important intermediate during biogas formation? There are no acetoclastic methanogens detected and there are no syntrophic acetate oxidizers detected! Hence, it is strikingly obvious that these primers give a misleading picture of the archaeal community. Consequently, when knowing this, it is not scientifically sound to present the archaeal community composition based on these primers. Hence, archaeal reads should be filtered out and only the bacterial community should be presented. And yes, there are a lot of studies who use two separate primer sets, one for bacteria and one for archaea (either targeting archaeal 16S rRNA genes or mcrA gene for example).

2.Our response: We fully agree with the reviewer that the proportion of methanogens is very low, which is unexpected since biomethane was produced during these experiments. The reviewer has suggested that we remove the data on methanogens, and concentrate on bacterial data only. While this acceptable, however, we would like to propose the following alternative that will take the reviewer’ comments into account. Since some methanogen archaeas have been detected, although be it at a very low proportion, we suggest to keep these data as they are, however, we have added a caveat that highlights clearly the limitation of our result. In this caveat (see page 21), we have clearly stated that the primers we used were not covering archaea sufficiently, hence the low level of archaea detection. Thus, with this information, we feel that the reviewer’ comment is fully addressed, while the same time, we are providing the readers with appropriate information showing the limitation of our results. 

3. Greengenes vs SILVA/MiDAS: I do not quite understand why the amplicon data analysis yields more counts, more OTUs when using Greengenes compared to SILVA. This does not make any sense. OTUs are constructed first and just after this step representative sequences of each OTU are taxonomically assigned by using one of the databases. Hence, OTU construction happens before the databases come into play and hence, is completely independent of the database. How then can the database influence the number of counts? And again Greengenes is outdate and should not be used anymore!

3.Our response: As the reviewer has suggested, we have reanalysed our data using the SILVA/MiDAS software. News figures have been made using the data generated from this SILVA/MiDAS (Figures 6 to 10), and the text has been changed to reflect all these new results (highlighted in yellow, from page 17-21). 

Overall, there is no major change on the result on microbial community and taxa, changes are minor, and have been incorporated in the text.

---

## [Decision Letter · Decision Letter 2]

28 Oct 2019

Characterisation and microbial community analysis of lipid utilising microorganisms for biogas formation.

PONE-D-19-15756R2

Dear Dr. Nzila,

We are pleased to inform you that your manuscript has been judged scientifically suitable for publication and will be formally accepted for publication once it complies with all outstanding technical requirements.

With kind regards,

Juan J Loor

Academic Editor

PLOS ONE

Additional Editor Comments (optional):

Reviewers' comments:

Reviewer's Responses to Questions

**Comments to the Author**

1. If the authors have adequately addressed your comments raised in a previous round of review and you feel that this manuscript is now acceptable for publication, you may indicate that here to bypass the “Comments to the Author” section, enter your conflict of interest statement in the “Confidential to Editor” section, and submit your "Accept" recommendation.

Reviewer #1: All comments have been addressed

2. Is the manuscript technically sound, and do the data support the conclusions?

Reviewer #1: Yes

3. Has the statistical analysis been performed appropriately and rigorously? 

Reviewer #1: Yes

4. Have the authors made all data underlying the findings in their manuscript fully available?

Reviewer #1: Yes

5. Is the manuscript presented in an intelligible fashion and written in standard English?

Reviewer #1: Yes

6. Review Comments to the Author

Reviewer #1: (No Response)

7. PLOS authors have the option to publish the peer review history of their article (what does this mean?). If published, this will include your full peer review and any attached files.

Reviewer #1: No

---

## [Editor Report · Acceptance letter]

31 Oct 2019

PONE-D-19-15756R2 

Characterisation and microbial community analysis of lipid utilising microorganisms for biogas formation. 

Dear Dr. Nzila:

I am pleased to inform you that your manuscript has been deemed suitable for publication in PLOS ONE. Congratulations! Your manuscript is now with our production department. 

With kind regards,

on behalf of

Dr. Juan J Loor 

Academic Editor

PLOS ONE